# A Preliminary Study on Factors That Drive Patient Variability in Human Subcutaneous Adipose Tissues

**DOI:** 10.3390/cells13151240

**Published:** 2024-07-24

**Authors:** Megan K. DeBari, Elizabeth K. Johnston, Jacqueline V. Scott, Erica Ilzuka, Wenhuan Sun, Victoria A. Webster-Wood, Rosalyn D. Abbott

**Affiliations:** 1Department of Biomedical Engineering, Carnegie Mellon University, Pittsburgh, PA 15213, USA; mdebari@andrew.cmu.edu (M.K.D.); elizabej@andrew.cmu.edu (E.K.J.); jacquels@andrew.cmu.edu (J.V.S.); eiizuka@andrew.cmu.edu (E.I.); vwebster@andrew.cmu.edu (V.A.W.-W.); 2Department of Mechanical Engineering, Carnegie Mellon University, Pittsburgh, PA 15213, USA; sunwh345@gmail.com

**Keywords:** adipose tissue, patient variability, obesity, inflammation

## Abstract

Adipose tissue is a dynamic regulatory organ that has profound effects on the overall health of patients. Unfortunately, inconsistencies in human adipose tissues are extensive and multifactorial, including large variability in cellular sizes, lipid content, inflammation, extracellular matrix components, mechanics, and cytokines secreted. Given the high human variability, and since much of what is known about adipose tissue is from animal models, we sought to establish correlations and patterns between biological, mechanical, and epidemiological properties of human adipose tissues. To do this, twenty-six independent variables were cataloged for twenty patients, which included patient demographics and factors that drive health, obesity, and fibrosis. A factorial analysis for mixed data (FAMD) was used to analyze patterns in the dataset (with BMI > 25), and a correlation matrix was used to identify interactions between quantitative variables. Vascular endothelial growth factor A (VEGFA) and actin alpha 2, smooth muscle (ACTA2) gene expression were the highest loadings in the first two dimensions of the FAMD. The number of adipocytes was also a key driver of patient-related differences, where a decrease in the density of adipocytes was associated with aging. Aging was also correlated with a decrease in overall lipid percentage of subcutaneous tissue, with lipid deposition being favored extracellularly, an increase in transforming growth factor-β1 (TGFβ1), and an increase in M1 macrophage polarization. An important finding was that self-identified race contributed to variance between patients in this study, where Black patients had significantly lower gene expression levels of TGFβ1 and ACTA2. This finding supports the urgent need to account for patient ancestry in biomedical research to develop better therapeutic strategies for all patients. Another important finding was that TGFβ induced factor homeobox 1 (TGIF1), an understudied signaling molecule, which is highly correlated with leptin signaling, was correlated with metabolic inflammation. Furthermore, this study draws attention to what we define as “extracellular lipid droplets”, which were consistently found in collagen-rich regions of the obese adipose tissues evaluated here. Reduced levels of TGIF1 were correlated with higher numbers of extracellular lipid droplets and an inability to suppress fibrotic changes in adipose tissue. Finally, this study indicated that M1 and M2 macrophage markers were correlated with each other and leptin in patients with a BMI > 25. This finding supports growing evidence that macrophage polarization in obesity involves a complex, interconnecting network system rather than a full switch in activation patterns from M2 to M1 with increasing body mass. Overall, this study reinforces key findings in animal studies and identifies important areas for future research, where human and animal studies are divergent. Understanding key drivers of human patient variability is required to unravel the complex metabolic health of unique patients.

## 1. Introduction

Long thought of as biologically inert, white adipose tissue is now recognized as a highly active endocrine organ that plays a critical role in systemic hormone, cytokine, immune, and metabolic regulation. Given its dynamic state and regulatory role in the body, changes in adipose tissue have profound effects on the health of patients. For example, accumulation of fat mass in obesity is a risk factor for a plethora of diseases from diabetes to certain cancers [1,2]. Furthermore, inconsistencies in human adipose tissues are extensive and multifactorial, including large variability in cellular sizes, lipid content, inflammation, mechanics, extracellular matrix components, and cytokines secreted [3,4,5,6]. This large patient variability leads to poor fat graft consistency and survival when the tissue is implanted [7,8,9,10] and inadequate treatment options for disorders related to adipose tissue dysfunction. For example, differences in patient demographics such as race and bariatric surgery have led to profound differences in insulin sensitivity between patients [11,12,13,14,15,16]. Therefore, determining key variables that drive the large variability in human adipose tissue structure and function is critical.

The predominant cell in adipose tissue is the adipocyte, which stores triglycerides that can be released into the blood stream as fatty acids when energy is required. Additional cells make up the stromal vascular fraction (SVF), which is composed of adipose-derived stem cells (ASCs), preadipocytes, immune cells, fibroblasts, pericytes, and endothelial cells [17]. White adipose tissue hosts a variety of resident and infiltrating immune cells, including T and B lymphocytes, macrophages, neutrophils, and eosinophils, that maintain tissue homeostasis [18].

From obese animal models, it is known that adipose tissue acts as a reservoir, buffering nutrient surplus with lipid storage and tissue expansion through adipocyte enlargement (hypertrophy) and formation of new adipocytes (hyperplasia) [19,20]. While adipocyte hypertrophy enables adipose tissue to store more lipids, it decreases the surface area-to-volume ratio, resulting in ineffective nutrient transport and poor cell signaling, promoting tissue dysfunction and metabolic disorders [21]. In fact, hypertrophic adipocyte growth correlates with diabetes in obese humans [12], whereas hyperplastic growth correlates with improved insulin sensitivity in human subjects [22].

To limit adipose tissue expansion, transforming growth factor-β1 (TGFβ1) acts as an inhibitor of adipogenesis [23] and drives fibrotic changes in adipose tissue [24]. Adipose tissue fibrosis is defined as the chronic exposure of adipose tissues to inflammation and hypoxia that leads to a state of extracellular matrix remodeling and collagen deposition [25]. Obesity is one of the main factors leading to fibrosis of adipose tissue [26,27]; however, not every patient that is obese will have fibrotic tissue. Fibrosis is a known contributor to varying patient responses to bariatric surgery [6,28].

Since much of what we know about adipose tissue is from animal models, in this study, we sought to establish correlations and patterns between biological, mechanical, and epidemiological properties of human adipose tissues through statistical analyses (Figure 1). Based on key roles in health, obesity, and fibrosis, 20 independent variables were chosen. The independent variables were paired with six categories of patient demographics (Table 1) that were available for each sample. Here, the ultimate goal was to establish key variables that drive patient variability and to determine if similar findings in human tissues are consistent with those of animal studies. For full demographic information on each patient, see Table 2.

## 2. Methods

### 2.1. Adipose Tissue Procurement

Discarded, deidentified adipose tissue (*n* = 20 different patients) was obtained from elective plastic surgery procedures at the University of Pittsburgh Medical Center (UPMC) with the Institutional Review Board (IRB No. 0511186). To randomly select patients, adipose tissue was requested only knowing the surgery type. All patient demographics available were used in analysis of this study, including gender, body mass index (BMI), diabetic status, smoking status, age, surgery the adipose tissue was collected from, previous weight loss strategy, and race (Table 1). Other information, such as comorbidities, was not provided for all patients, so this information was excluded from this study, but it would be advantageous to include in the future. It is important to note that while the patient’s diabetic status was provided, the type (type I or type II) was not disclosed. Additionally, because the quantity, duration, and quit date were not provided for patients identified as former smokers, they were grouped with current smokers. This grouping was done to account for residual or long-lasting effects of smoking. Future work should explore this variable further. Adipose tissue was procured from the Adipose Stem Cell Research Center at the University of Pittsburgh’s Medical Center’s Department of Plastic Surgery. Due to the availability of samples, only overweight and obese patients were obtained (BMI > 25). Subcutaneous adipose tissue samples were dissected from abdominal fat taken from either an abdominoplasty or a panniculectomy surgery. A bulk sample with a minimum size of 4” by 4” was used for each experiment. For all testing, samples were taken from directly below the fascia of scarpa.

### 2.2. Imaging

Tissue samples were isolated from bulk tissue in a 3 × 3 grid. The tissue samples were fixed in formalin (Sigma-Aldrich, St. Louis, MO, USA). Six of the samples were randomly selected and stained with AdipoRed (Lonza, Walkersville, MD, USA) (1:35) and Alexa Fluor Phalloidin 488 (Thermo Fisher Scientific, Waltham, MA, USA) (1:100) and imaged using confocal and multiphoton microscopy (Nikon, Melville, NY, USA) to visualize lipids, f-actin, and collagen via Second Harmonic Generation. Images taken using confocal microscopy were used to determine the size of extracellular lipid droplets. Two samples per patient were imaged, with two images being captured per sample, for a total of four images per patient and 80 images total for the whole study. A total of 1904 extracellular lipid droplets were measured. Extracellular lipid droplets were defined as lipids that were not intracellularly contained, as indicated by the exclusion from the cytoskeletal Phalloidin staining. The mean diameter was determined to be 7.27 μm, with a standard deviation of 8.16 μm (Figure 2). To encompass a majority of extracellular lipid droplets, the mean plus one standard deviation (15.43 μm) was determined to be the maximum size of what we defined as an extracellular lipid droplet. This accounted for 84.1% of extracellular lipid droplets in the samples imaged and was well below the smallest adipocytes, which are >20 μm [3]. Every patient and field of view had extracellular lipid droplets; however, the number of lipid droplets varied slightly from patient to patient. 

The extracellular and intracellular lipid droplets and collagen were visualized with multiphoton microscopy. Depending on the sample and the location of the image, the collagen to adipocyte distribution varied. To account for the high variability of collagen/adipocyte distribution, many images were captured and analyzed. Six samples were imaged, with three images being captured per sample, for a total of 360 images. These images were used to evaluate (1) adipocyte diameter; (2) the extracellular lipid percent; and (3) whether extracellular lipids were in high-density collagen areas (fibrotic regions). Only lipid droplets that could be clearly measured were used for analysis. The number of adipocytes and extracellular lipid droplets were counted and used to determine the extracellular lipid percent, defined as
$$Extracellular\;lipid\;percent=\frac{{total}_{Extracellular\;lipid\;droplets}}{{total}_{Extracellular\;lipid\;droplets}+{total}_{Adipocytes}}$$
where *total_Extracellular lipid droplets_* represents the total number of extracellular lipid droplets in each image, and *total_Adipocytes_* represents the total number of adipocytes counted in each image. Additionally, the location of the extracellular lipid droplets was categorized by their proximity to collagen. Extracellular lipid droplets embedded in or touching collagen fibrils were identified as “extracellular lipid droplets in collagen”. This was used to determine if extracellular lipid droplets were co-localized within the collagen: $$Co-localization\;Score=\frac{number\;of\;extracellular\;lipid\;droplets\;in\;collagen}{total\;number\;of\;extracellular\;lipid\;droplets}\;$$

All image analysis measurements were done using ImageJ v1.53k. An average of 528 adipocytes were measured per patient to obtain diameter metrics. All images were taken using the same magnification. For the full set of quantification results for each patient, see Appendix A.

### 2.3. DNA Content

PicoGreen assays (Thermo Fisher Scientific, Waltham, MA, USA) were performed on each patient’s tissue samples following the manufacturer’s procedure to assess DNA content. The DNA content was normalized to the weight of each tissue sample. *n* = 5 for each patient. Samples were run in technical duplicates.

### 2.4. Metabolic Activity/Redox Readout

Resazurin (Thermo Fisher Scientific, Waltham, MA, USA) was diluted to 1 mM with phosphate buffered saline (PBS) (pH 7.4). This was diluted further to 0.05 mM solution using cell culture media (DMEM with 10% FBS and 1% Pen-Strep). An amount of 1 mL of the 0.05 mM resazurin/media solution was placed in a 24-well plate with the tissue samples. The samples were incubated at 37 °C for 2.5 h. Using a microplate reader (SpectraMax i3x), the absorbance at 570/600 nm was measured. Absorbance data were normalized to the average DNA content per gram of tissue for each patient. *n* = 5 for each patient. Samples were run in technical duplicates.

### 2.5. Collagen Content

Hydroxyproline assays (Sigma-Aldrich, St. Louis, MO, USA) were performed on each patient’s tissue samples following the manufacturer’s protocol to assess collagen content. Briefly, approximately 10 mg was homogenized in 100 µL of water (exact tissue weight was recorded before homogenization). A total of 100 µL of 12 M HCL was added and hydrolyzed for 3 h. The supernatant was then transferred to a well plate and allowed to evaporate. Chloramine T/Oxidation Buffer and Diluted DMAB Reagent were then added following the manufacturer’s protocol. Collagen content was normalized to the weight of the initial tissue sample. *n* = 1 for each patient. Samples were run in technical duplicates.

### 2.6. Advanced Glycation End Products (AGE)

AGE assays (Abcam, Cambridge, MA, USA) were performed on each patient’s tissue samples. Tissue samples were flash frozen in liquid nitrogen and stored at −80 °C until use. Tissue samples were then suspended in PBS (pH 7.4) (100 mg of tissue to 1 mL of PBS), homogenized, and stored at −20 °C overnight. The following day, homogenized tissues were subjected to two freeze–thaw cycles. An amount of 1 mL of the solution was then centrifuged at 5000× *g* at room temperature for 5 min. Samples for this assay were taken from right below the lipid layer of the centrifuged homogenized tissue to avoid the lipid layer. The assay was performed following the manufacturer’s procedure. *n* = 1 for each patient. Samples were run in technical duplicates.

### 2.7. Compression Testing

Tissue samples were compressed to 90% strain at a rate of 1 mm/min using an electromechanical universal testing system (MTS Criterion, MTS, Eden Prairie, MN, USA). The elastic modulus was determined by the linear portion of the stress strain curve between 60 and 90% strain. The adipose tissue samples had stress-strain curves similar to elastomers [29]. The stress remained low until around 50–60% strain, when it began to increase drastically. A strain range of 60–90% was used to determine the moduli because this is the region of the curve where the extracellular matrix proteins (ECM) are being compressed. It is thought that prior to the drastic increase in stress, adipocytes are being compressed but not to failure. Once the adipocytes are compressed to failure, the stress increases quickly. This is further supported visually by the lipids or “oil” being seen pooling around the sample as the stress began to increase. Each sample was compressed to 90% strain, and at this point, the peak stress was recorded. *n* = 10 for each patient. 

### 2.8. Stromal Vascular Fraction (SVF) Doubling Time

The SVF was isolated from subcutaneous adipose tissue as described previously [30]. The cells were isolated by mechanically blending the adipose tissue until the texture resembled lipoaspirate. The tissue was then incubated in a collagenase solution (0.1% collagenase, 1% bovine serum albumin, 98.9% phosphate buffer solution) (Thermo Fisher Scientific, Waltham, MA, USA) in a 1:1 ratio for 1 h at 37 °C. Following the incubation, the solution was centrifuged (5 min at 300× *g*) to isolate the SVF. The cells were resuspended in media (DMEM with 10% FBS and 1% pen-strep), centrifuged, and seeded into flasks. Once cells reached confluency, cells were trypsinized, and 10,000 cells/well were seeded into a 6-well plate to ensure comparable log-phase growth. Media was changed every 2–3 days. A total of 3 images of each well were taken every other day until confluency was reached. Cells were counted, and the doubling time was calculated using the following equation:$$DT=(t_{2}-t_{1})\frac{ln(2)}{ln(\frac{q_{2}}{q_{1}})}$$
where *DT* = doubling time, *t_2_* = second time point, *t_1_* = first time point, *q_2_* = number of cells at second time point, and *q_1_* = number of cells at first time point. *n* = 5 for each patient. Due to a COVID-19 laboratory shutdown, patients 5 and 6 are missing doubling time data. For the complete set of data for each patient related to mechanical properties, collagen content, AGEs, metabolic activity, and doubling rate, see Appendix A.

### 2.9. Reverse Transcription Quantitative Real-Time Polymerase Chain Reaction (RT-qPCR)

Tissue samples were flash frozen in liquid nitrogen and stored at −80 °C until RNA isolation. Total RNA was isolated using the RNeasy Lipid Tissue Mini Kit (QIAGEN, Venlo, Netherlands), and concentration was measured using the NanoDrop 2000c spectrophotometer (Thermo Scientific, Waltham, MA, USA). RNA was processed into cDNA using the iScript cDNA Synthesis Kit (Bio-Rad, Hercules, CA, USA). qPCR reactions were performed with 100 ng of cDNA per reaction. Succinate dehydrogenase complex flavoprotein subunit A (SDHA) (ID: 6389) was used as a reference housekeeping gene [31] and interleukin 6 (IL6) (ID: 3569), tumor necrosis factor α (TNFα) (ID: 7124), TGFβ induced factor homeobox 1 (TGIF1) (ID: 7050), leptin (LEP) (ID: 3952), adiponectin (ADIPOQ) (ID: 9370), transforming growth factor β 1 (TGFβ1) (ID: 7040), actin alpha 2, smooth muscle (ACTA2) (ID: 59), CD163 (ID: 9332), CD86 (ID: 942), and vascular endothelial growth factor A (VEGFA) (ID: 7422) were included as genes of interest. All patient and primer combinations were performed with technical duplicates. The CT value was determined through regression analysis performed by the Bio-Rad CFX96 Real-Time System and C1000 Touch Thermal Cycler (Bio-Rad, Hercules, CA, USA). Since there was no control group, only ΔCT was calculated:ΔCT = CT_SDHA_ − CT_Target gene_

Using this formula, higher CT values of the target gene, which indicate lower gene expression, would have lower ΔCT values. A CT of 45 was assigned for genes that were not detected after 40 cycles. For the full set of patient PCR results, see Appendix A.

### 2.10. Statistics

Quantitative variables were evaluated for differences between patients using either a one-way ANOVA or two-way ANOVA followed by a Tukey’s post-hoc analysis (GraphPad Prism 9.0.0). A one-way ANOVA was used to determine statistical differences between adipocyte diameters, number of extracellular lipid droplets in collagen/total number of extracellular lipid droplets co-localization, elastic moduli, peak stress, doubling time, and metabolic activity. A two-way ANOVA was used to determine statistical differences between the percentages of pixels (collagen/lipids). Comparisons between ancestry of the cells (self-identified “White” versus “Black”) were performed with a *t*-test. *p* values less than 0.05 were determined to be statistically significant. All data are represented as mean +/− standard deviation. A factorial analysis for mixed data (FAMD) was performed on the data using R statistics software. To get all of the variables on the same scale, data were standardized before conducting the analysis by subtracting the mean of the dataset from every value and then dividing by the standard deviation [32,33]. Because complete datasets are required to perform an FAMD, the missMDA package was used to estimate the two missing doubling time data points [34] from a COVID-19 lab closure. The FAMD analysis was performed using the FactoMineR package, with additional graphs being created with the factoextra package. Cos2 values were normalized to the highest value in each dimension to determine significant correlations. Coordinates for quantitative and qualitative data for each dimension are included in Appendix A. R-code used to generate FAMD results is included in Appendix A. Using the GraphPad Prism statistical software, all quantitative data were used to generate a correlation matrix using the Pearson correlation coefficient with a confidence interval of 95%.

## 3. Results and Discussion

Dimensionality reduction is used to identify relationships amongst variables and extract patterns in datasets. To determine if there were relationships between variables in our dataset, we used two different statistical models: (1) a correlation matrix to evaluate relationships between quantitative variables (Figure 3) and (2) a factorial analysis of mixed data (FAMD) to understand the relationships between qualitative (race, sex, surgery type, diabetic status, smoking status) and quantitative variables. An FAMD is similar to principal component analysis (quantitative data only) and multiple correspondence analysis (qualitative data only), but both quantitative and qualitative data can be analyzed [34]. With the FAMD, the number of dimensions needed to account for > 80% of the variability in our dataset was determined to be nine (Table 3; for full set of coordinates, see Appendix A). The largest loading was the variable that contributed the most variance in the given dimension, meaning that variability in these key factors was correlated with variability in the rest of the dataset and is an essential contributor to patient-related differences. 

### 3.1. Levels of Vascular Endothelial Growth Factor A (VEGFA) Gene Expression Accounted for the Most Variance in the Full Dataset of Variables (Qualitative and Quantitative)

The variable that contributed the most to the first dimension of the FAMD analysis was gene expression of VEGFA (Table 3). This means that variability in VEGFA gene expression was linked to the most patient-related differences in the other variables in this study. In all individuals, the resident vasculature of adipose tissue acts to ensure adequate blood flow and nutrient/waste transport to allow for adipose tissue expansion and metabolism. In adipose tissue, VEGFA is a pleiotropic molecule that is involved in vasculogenesis, angiogenesis, vascular permeability, tissue remodeling, and metabolic effects (see our recent reviews: [35,36]). Mouse models that overexpress VEGFA result in an increase in adipose tissue vascularization, systemic protection against metabolic dysfunction from a high-fat diet, and higher energy expenditure [37,38]. In the patient population evaluated here, higher gene expression of VEGFA was positively correlated with a longer doubling time of seeded stromal vascular fraction cells (0.60) and higher adiponectin gene expression (0.73) (Figure 4). 

VEGFA is widely expressed in adipose tissue by both the SVF as well as the mature adipocytes [39]. In our study, gene expression was taken from the whole tissue; therefore, the source of the VEGFA signal could be from the vasculature and/or the adipocytes. When the SVF is expanded in vitro, the predominant cells that persist are multilineage ASCs [40]. Therefore, the doubling time in our study reflects the proliferation of the ASCs. Many interacting factors can impact the proliferative capacity of ASCs, for example, diabetes [41], aging [42,43], and certain drugs [44]. However, the correlation between VEGFA and longer ASC doubling time has not been reported. It is well accepted that while proangiogenic, the exogenous application of VEGFA on endothelial cells can have a preferential response to either proliferation or migration depending on the need [45]. One possibility is that similar to endothelial cells, ASCs may have a more migratory response to support angiogenesis. Additionally, in adipose tissue, the balance between stem cell self-renewal and differentiation must be maintained. The regulatory mechanisms for maintaining the stem cell state as well as fate determination are highly conserved. With how heavily intertwined angiogenesis and adipogenesis are, it is not surprising that VEGF is one of these regulators, as blocking VEGFR2, preadipocyte differentiation is also inhibited [46]. Since VEGFA is increased during adipocyte differentiation [46], the positive correlation between VEGFA and a longer doubling time could indicate that the individual is experiencing more hyperplastic adipose tissue growth where ASCs are recruited and differentiate towards adipocytes rather than undergoing proliferation. Supporting this theory, a longer SVF doubling time was also correlated with a higher number of adipocytes (Appendix A, 0.42). Future work should characterize what cells are modulating VEGFA gene expression (the SVF and/or adipocytes) and the relationship between VEGFA and doubling time/adipocyte differentiation. Since endothelial cells represent 10–20% of the SVF [47], future investigations should also evaluate whether the endothelial cell percentage in the SVF affects the overall doubling time of the SVF.

The strong correlation between adiponectin and VEGFA supports findings that adiponectin is down-regulated in VEGFA ablated mice [48,49]. Adiponectin is often considered to be a proangiogenic factor [50,51]. In certain diseases such as chondrosarcoma [52] and rheumatoid arthritis [53], adiponectin promotes VEGFA-dependent angiogenesis. However, other studies have suggested that adiponectin induces endothelial cell apoptosis, reduces their proliferative capacity [54], and inhibits endothelial migration [55]. Therefore, further research is required to discern the correlation observed between adiponectin and VEGFA in the patient population evaluated here. VEGFA is also known to have a protective role in adipose tissue inflammation by recruiting M2 macrophages to adipose depots in mice [56]. Supporting this finding, our dataset indicated there was a low correlation between VEGFA and CD163 (0.36), with the FAMD indicating that these factors were clustered (Appendix A). Overall, these findings support a role of adiponectin and M2 macrophage recruitment in VEGFA-mediated metabolic protection.

### 3.2. TGFβ Induced Factor Homeobox 1 (TGIF1) Gene Expression Was a Key Contributor to the Highest Loading Direction

The second largest contributor in the first dimension of the FAMD was gene expression values for TGIF1 (Figure 5). The role of TGIF1 in adipose tissue has only recently been uncovered. TGIF1 represses TGFβ signaling via multiple pathways, including (1) direct Smad2 inhibition forming a transcriptional repressor complex, (2) prevention of Smad2 phosphorylation, or (3) targeting Smad2 for ubiquitin-dependent degradation [57,58,59]. In preadipocytes, insulin antagonizes TGFβ signaling in preadipocytes through TGIF1 transcription and is required for the differentiation of preadipocytes [60]. In fact, a recent screening study identified lower TGIF1 transcription as a key reducer of lipid accumulation in differentiating stem cells [61]. TGIF1 positively interacted with leptin, IL6, CD163, CD86, and no bariatric surgery, while negatively interacting with the number of extracellular lipid droplets (Figure 6).

Leptin is a pleiotropic hormone predominantly synthesized by adipose tissue that is produced in greater quantities with increasing adipose tissue mass [62]. While leptin is known to stimulate liver fibrosis [63,64,65], it has been shown to inhibit fibrosis in mouse adipose tissue [66,67]. Our data suggest that TGIF1 could have a role in leptin suppression of fibrosis, as there was a high correlation between higher gene expression levels of TGIF1 and leptin (0.71). It is possible that an increase in leptin stimulates higher TGIF1, suppressing TGFβ-triggered fibrotic changes. However, given the endpoint nature of the study, causation cannot be determined. Therefore, determining the directionality of the leptin and TGIF1 interaction will be an important next step. Another striking correlation is that lower levels of TGIF1 are correlated with higher numbers of extracellular lipid droplets (−0.48). Further work needs to explore the role of extracellular lipid droplets observed in adipose tissue and their association with TGIF1.

Higher levels of TGIF1 were positively correlated with interleukin-6 (Il6) and macrophage surface markers CD163 and CD86 (Figure 6B–D, respectively), suggesting that high levels of TGIF1 expression could have a role in metabolic inflammation in adipose tissue. Adipocyte secretion of IL6 increases macrophage infiltration in adipose tissue [68] and is linked with reduced insulin sensitivity and diabetes [69]. While both cell surface markers are present on polarized macrophages, CD86 is typically considered to be a classically activated proinflammatory M1 macrophage marker, and CD163 is an alternatively activated anti-inflammatory M2 macrophage marker [70,71]. In the overweight/obese population studied here, CD86 and CD163 were positively correlated (0.61). It is not unusual to see high levels of both CD86 and CD163 gene expression in adipose tissue samples with BMI > 25, as others have noted this signature as indicative of metabolic inflammation [71]. Furthermore, TGIF1 has been identified as a potential transcriptional regulator in macrophage activation [72].

The correlation between TGIF1 gene expression and bariatric surgery is interesting. Bariatric surgery results in rapid weight loss and is an effective treatment to restore insulin sensitivity and reduce type 2 diabetes complications [73,74]. Our results indicated that in some patients, TGIF1 levels of patients who had previously undergone bariatric surgery were well below levels of those who had not (Figure 6F). This is an important finding suggesting that some patients that undergo bariatric surgery will have low levels of TGIF1 and will likely not have TGIF1-mediated TGFβ1 signaling suppression. However, lower levels of TGIF1 were not seen in all patients who underwent surgery, suggesting that other variables also interact.

It is not surprising that TGIF1 and TGFβ1 gene expression levels were not correlated in our adipose tissue samples (0.12). This is because TGIF1 acts downstream from TGFβ1, repressing the transcriptional response [57] rather than affecting the expression of the TGFβ1 gene. This is seen in other tissues as well. For example, in the glomerulus, the profibrotic action of TGFβ1 is antagonized by TGIF and blocks α-smooth muscle activation; however, high levels of TGIF do not affect TGFβ1-mediated Smad2 phosphorylation and its nuclear translocation [75].

### 3.3. ACTA2 Was the Second Loading Dimension of the FAMD

ACTA2 contributed the most to the second dimension of the FAMD. ACTA2 is a gene that encodes smooth muscle alpha (α)-2 actin production, which is required for cell movement and muscle contraction. ACTA2 (Figure 7) was correlated with TGFβ1 (0.69) and collagen content (0.58), while being inversely related to adipocyte diameter (−0.39). This is consistent with studies where murine adipocytes exposed to a high fat diet upregulate expression of extracellular matrix genes and ACTA2 gene expression characteristic of a myofibroblast-like cell type through reduced PPARγ activity and elevated TGFβ-SMAD signaling [76,77]. The ACTA2 “cellular identity crisis” is thought to drive functionality changes in obese adipose tissues [76], where it is known that enhanced collagen deposition restricts adipocyte size [78]. This was further supported by our dataset’s low inverse correlation between collagen percent and adipocyte diameter (−0.32). Collectively, our data suggests a strong role for ACTA2 signaling as a contributor to the variability in human adipose tissues, where high ACTA2 gene expression is related to fibrosis (collagen deposition and TGFβ1) and a reduction in adipocyte cellular diameter.

### 3.4. TGFβ1 Expression

Besides being correlated with ACTA2 (0.69), as discussed in the last section, TGFβ1 was correlated with hydroxyproline content (0.54), the classically activated M1 macrophage surface marker gene expression (CD86, 0.46), and VEGFA gene expression (0.42). These results are consistent with the known fibrotic role of TGFβ1 in adipose tissue characterized by collagen accumulation, hypoxia (which upregulates VEGFA [79]), and M1 macrophage polarization [25,79,80,81,82].

### 3.5. Age-Related Differences

In our dataset, aging was correlated with a decrease in the number of adipocytes (Figure 8A, −0.549) and percentage of lipids (Figure 8B, −0.601). It is important to note that age and adipocyte size had no correlation (−0.078), indicating that the observed decrease in the number of adipocytes with age was not due to an increase in the size (hypertrophy) of adipocytes. Instead, our findings are consistent with other studies that have shown that hyperplastic growth in adipose tissue declines with age [21]. Furthermore, the decrease in lipid percentage with age and the lack of correlation between BMI and percentage of lipids (−0.186) in our subcutaneous adipose tissue samples support studies that show a shift in lipid storage from subcutaneous to visceral depots with age [62,63]. The ability to buffer lipids in adipocytes declines with age, leading to ectopic lipid deposition in the liver and muscle [83]. Supporting the shift in adipocyte lipid storage capacity, there was an increase in the number of extracellular lipid droplets with age (Figure 8D, 0.370).

As mentioned previously, aging was correlated with an increase in TGFβ1 gene expression (Figure 8C, 0.40) and upregulation of a M1 macrophage marker (CD86, 0.33) while displaying no correlation with a M2 macrophage marker (CD163, 0.05). It is known that aging results in compounded reactive oxygen species, which readily primes macrophages toward a proinflammatory M1 state and promotes age-related metabolic syndrome (atherosclerosis, obesity, and type II diabetes) [84]. TGFβ signaling largely depends on the cell type and context. However, the upregulation of TGFβ ligands with age is known to contribute to cell degeneration, tissue fibrosis, inflammation, decreased regeneration capacity, and metabolic malfunction in other tissues [85]. Therefore, age-associated changes in subcutaneous adipose tissue are an important context that warrants further investigation.

### 3.6. Ancestral Differences

Race was the highest loading in the fourth dimension, which accounted for over 11% of the variance in the dataset. Despite known differences in disease prevalence in diverse patient populations, cellular ancestry is largely overlooked in biomedical research [86]. This ancestry-blind approach ignores important differences in disease progression that are key to better therapeutic options, such as ancestral differences in cellular transcription [87] and drug responsiveness [88]. Adipose tissue plays a central role in obesity and type II diabetes progression, with many studies indicating a greater prevalence in Hispanic and Black patients over White patients [89,90,91].

In the current study, samples derived from self-identified Black patients had significantly lower gene expression levels of TGFβ1 and ACTA2 (Figure 9A,B) and a trend towards lower hydroxyproline collagen content (Figure 9C) than White patients (all patients self-identified as non-Hispanic). As mentioned previously, fibrosis is characterized by enhanced collagen deposition, driven by high levels of TGFβ1 and its target gene ACTA2, which induces a myofibroblastic phenotype [25,77]. TGFβ signals resolution of inflammatory signals, and therefore attenuated TGFβ/Smad signaling results in a reduced mechanism to resolve inflammation [92]. The observed low fibrotic signature in adipose tissues samples derived from Black patients is consistent with Black patients having lower incidences of pulmonary fibrosis than non-Hispanic White patients [93]. Therefore, future work should explore the role of ancestry in subcutaneous adipose tissue fibrosis.

Our data support the concept that disease progression could present differently in different patient populations. For example, there is an observed trend that Hispanic patients disproportionally develop nonalcoholic fatty liver disease (NAFLD) and have high rates of obesity, visceral adipose tissue, and insulin resistance. In contrast, Black patients tend to have a high prevalence of obesity and insulin resistance with a paradoxically favorable lipid profile and low prevalence of visceral adipose tissue and NAFLD [94]. On average, samples from Black patients in our study had higher gene expression levels of TGIF1 (Figure 9D) and leptin (Figure 9E) with fewer extracellular lipid droplets (Figure 9F) compared to samples from White patients. We speculate that there could be a metabolic phenotype in some patients that favors intracellular lipid accumulation with high TGIF1 transcription that limits extracellular lipid release and (at least initially) favors a diabetic disease process over disease processes that are driven by lipid accumulation in other tissues. This is consistent with a recent study indicating that plasma triglyceride concentrations are lower in African Americans compared to non-Hispanic White, Hispanic White, East Asian, and South Asian ethnicities (controlling for the level of insulin sensitivity) [95]. Furthermore, preliminary findings from another research group indicated that TGIF1 overexpression leads to severe hyperglycemia and obesity-associated diabetes [96], suggesting a link between high TGIF1 signaling and an increased likelihood of developing a diabetic phenotype. The FAMD analysis also identified an interaction between sample ancestry and adiponectin gene expression. While not significant (by an unpaired *t*-test), lower levels of adiponectin were observed in samples derived from Black patients. Adiponectin is an insulin-sensitizing, anti-inflammatory, and anti-diabetic adipokine of which the levels decrease with increasing adipose tissue mass [97,98,99]. Therefore, lower adiponectin levels (with similar BMI ranges, Figure 9H) further support the higher observed rates of insulin resistance in Black patients compared to White patients [89,90,91]. Therefore, it is imperative that future work explores TGIF1 signaling, lipid release mechanisms from adipocytes, diabetic disease progression, and mechanisms of inflammation resolution in cells derived from diverse backgrounds.

Interestingly, the outliers in TGIF1, leptin, hydroxyproline, and number of extracellular lipid droplets were all the same patient (red dots). This further supports the aggregated data (with combined ancestries) that indicate that reduced levels of TGIF1 are linked with higher numbers of extracellular lipid droplets and an inability to suppress fibrotic changes in adipose tissue. It should also be noted that our sample size is small for this study, and we had uneven sample sizes for each group, with only six samples from self-identified Black patients and 14 from self-identified White patients. However, even with the small sample size from Black patients, the results are independent of BMI, with a similar spread of BMIs between the groups (Figure 9H). Collectively, these results underscore the urgent need to account for patient ancestry in biomedical research.

### 3.7. Macrophage Surface Markers

Macrophages play a major role in tissue homeostasis and disease, modulating the growth of tissues as well as tissue remodeling and organization. As mentioned previously, CD86 is considered a classically activated M1 macrophage marker, and CD163 is an alternatively activated M2 macrophage marker [70,71], and both were positively correlated in our dataset (Figure 10A). While macrophage polarization results in distinct functional phenotypes that are often thought of as divergent (i.e., an increase in M1 polarization is associated with a decrease in M2 polarization), induction routes are complex, interconnecting network systems rather than simple switches in activation patterns [100]. While oversimplifying the complexity of macrophage populations, it is well established that lean adipose tissue contains resident macrophages that are alternatively activated (M2) and maintain tissue homeostasis [85]. As adipose tissue increases in mass, there is infiltration of additional macrophages, and the number of classically activated (M1) macrophages increases, contributing to the inflammatory signature of obesity and insulin resistance [101,102], where there is an observed phenotypic switch from one activation state to another [103,104]. However, there is growing evidence that macrophage activation patterns within adipose tissue are more complicated than a full switch from M2 to M1 in obesity where both populations co-exist [105,106], which could be related to significant differences in macrophage profiles in animals and human models [107]. As mentioned previously, in humans, high levels of both CD86 and CD163 gene expression have been identified in samples from overweight/obese patients [71]. Consistent with this finding, both CD86 and CD163 also showed a high correlation with leptin in our dataset (0.60 and 0.64, respectively), a hormone that is directly related to total body fat mass in humans [41]. Therefore, our results indicate that higher levels of both M1 and M2 polarization markers are present in human subcutaneous adipose tissue with increasing body fat mass. Future work should evaluate the spectrum of polarization states in obese human tissues, including subtype-specific adipose tissue macrophage surface markers [108], to address the complexity of human macrophage populations [109].

Other correlations that were consistent for both macrophage surface markers, CD86 and CD163, were adipocyte diameter (0.46 and 0.33, respectively) and TGIF1 (mentioned previously and shown in Figure 6). Hypertrophic adipocytes are known to become inflammatory and necrotic, attracting M1 macrophages, which organize into “crown-like” structures that eventually clear necrotic adipocytes [110,111,112]. M2 macrophages regulate adipose progenitors by initiating proliferation and differentiation into adipocytes to mitigate existing adipocytes’ nutritional overload and prevent their further enlargement [113]. Therefore, it is unsurprising that both cell types correlate with hypertrophic adipocytes.

While the macrophage surface markers were correlated in our dataset, there were also distinct correlations associated with each polarization phenotype. For example, higher M1 CD86 gene expression was correlated with TGFβ1 (0.46 versus 0 for CD163), increasing age (0.33 versus 0.05 for CD163), and metabolic activity (−0.41 versus 0.14 for CD163), as discussed previously. Meanwhile, higher M2 CD163 gene expression was correlated with a higher overall lipid percent (0.45 versus 0.18 for CD86), which is likely related to the homeostatic role of M2 macrophages in recruiting the differentiation of new adipocytes [113].

### 3.8. Diabetic Status and AGEs

The largest loading in dimensions 6 and 7 of the FAMD was diabetic status (Table 3; combined, this accounts for ~12% of the variance). Diabetic status was clustered with AGEs and DNA content for dimensions 6 and 7, respectively (Figure 5). The largest loading in the final dimension of the FAMD (that was included to account for >80% of the variance) was AGEs and was clustered with diabetic status in that dimension as well. The AGE content in the tissue samples varied considerably (Appendix A). However, patients with diabetes or who formerly had diabetes had higher levels of AGEs. Patients with a higher BMI also had higher levels of AGEs. This trend is supported by literature that shows that increased AGE content in tissue is associated with age, diabetes, and obesity-related complications [114,115].

### 3.9. Smoking Status

The patients’ smoking status was the largest loading in the eighth dimension of the FAMD (Table 3, Figure 5). Adipose tissue has nicotinic receptors [116] that enable tobacco to greatly affect adipose tissue properties and cause downstream effects. The FAMD indicated that diabetic status and smoking were clustered. Diabetic patients who smoke have health problems, comorbidities, and poor outcomes compared to diabetic patients who do not smoke [117,118,119,120].

### 3.10. Limitations

The correlative nature of our investigation limits the conclusions that can be drawn in terms of cause and effect. Instead, we view this study as complementing current literature and informing future research directions into cause-and-effect relationships. It is also important to acknowledge the parameters of this study. The tissue samples used in this study were all from patients who were either overweight or obese (BMI > 25). Therefore, the correlations drawn from the data are not applicable to lean patients. Furthermore, tissue was taken from the subcutaneous adipose depot, and the results are therefore not applicable to other adipose depots in the body. However, it should be noted that there were no significant differences between tissues obtained from abdominoplasty and panniculectomy surgeries. All gene expression values were also taken from the bulk tissue; therefore, cell-type specific gene expression is not possible. For example, an upregulation of ACTA2 could mean that there are more myofibroblasts or that adipocytes are increasing their expression of the gene. As is the case for most studies, this preliminary study of 20 patient samples would be strengthened by the addition of more patients and an increase in the diversity of the patient population. For example, no patient samples were collected from Hispanic backgrounds. Therefore, replicating this study in locations with access to different patient demographics is essential. Assessing the differentiation capacity of ASCs would also have added value to this study and will be considered in future work. Additionally, more information on comorbidities, fat mass (which is more meaningful than BMI), genetics, and other lifestyle information such as history of exercise, diet, and smoking duration and cessation would strengthen and expand these results, providing valuable information for disease treatment and modeling.

## 4. Conclusions

Many of the results in this study agree with well-established findings in animal studies and other in vitro model systems. For example, TGFβ1 was associated with adipose tissue fibrosis, which is characterized by collagen accumulation and M1 macrophage polarization [25,27,80,81,82], and diabetes was associated with AGEs [114,115]. This study also revealed new patterns of key markers that drive human variability in adipose tissues.

VEGFA and ACTA2 gene expression were the highest loadings in the first two dimensions of the FAMD, respectively. Vascularization of human adipose tissue varies considerably between patients, and in this study, VEGFA was correlated with adiponectin gene expression. ACTA2 induces a so-called “cellular identity crisis”, as described elsewhere [76], which we found was correlated with higher collagen content and TGFβ1 signaling, a reduction in adipocyte cellular diameter, and an upregulation in VEGFA gene expression. There was also a key role of TGIF1 in accounting for variability in the adipose tissues, which has only recently been uncovered. This study emphasizes the importance of TGIF1 signaling for driving patient differences with a strong correlation to leptin signaling.

Furthermore, this study draws attention to what we define as “extracellular lipid droplets”, which were consistently found in the obese adipose tissues evaluated here. Reduced levels of TGIF1 were correlated with higher numbers of extracellular lipid droplets and an inability to suppress fibrotic changes in adipose tissue. We speculate that there could be a metabolic phenotype in some patients that favors intracellular lipid accumulation with high TGIF1 transcription that limits extracellular lipid release. High values for the colocalization score signify that extracellular lipid droplets were primarily found in collagenous regions of the subcutaneous adipose tissues (consistent across all patients, as seen in Appendix A). Further work needs to define what initiates secretion/deposition of lipid droplets extracellularly and whether extracellular lipid droplets have a role and/or association with the progression of adipose tissue fibrosis.

The number of adipocytes was the highest loading in the third dimension of the FAMD, where a decrease in the density of adipocytes was associated with aging and an increase in cellular proliferative capacity of ASCs. Aging was also associated with a decrease in overall lipid percentage that favored extracellular lipid droplet deposition, an increase in TGFβ1, and an increase in M1 macrophage polarization.

The highest loading in the fourth dimension of the FAMD, which accounted for over 11% of the variability in patient-related differences in this study, was self-identified race. While our sample size was small, Black patients had significantly lower gene expression levels of TGFβ1 and ACTA2. This finding indicates there is an urgent need to account for patient ancestry in biomedical research to develop better therapeutic strategies for all patients.

Amongst many other findings, this study indicated that M1 and M2 macrophage markers were correlated with each other and leptin (for BMI > 25). This finding supports growing evidence that macrophage polarization in obesity involves a complex, interconnecting network system rather than a simple switch in activation patterns from M2 to M1 with increasing body mass. Therefore, the characterization of M2 macrophages in overweight/obese tissues warrants further investigation.

## Figures and Tables

**Figure 1 cells-13-01240-f001:**
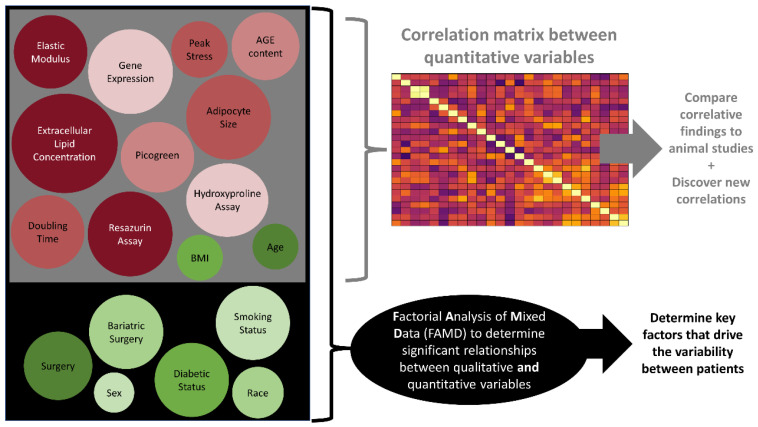
Study design showing how patient demographics (green) and tissue characterization techniques (red) fed into two different statistical models, where (1) each quantitative variable (gray box) was plotted in a correlation matrix against the other quantitative variables to determine correlative relationships and (2) qualitative and quantitative variables (all of the variables in the black box) were fed into a factorial analysis of mixed data (FAMD) to determine significant relationships between the variables. Correlations were used to compare the human data to published data from animal models (i.e., in animals, TGFβ1 gene expression is related to enhanced collagen deposition, and is verified in this study in human samples) and to discover new correlations (i.e., there is a correlation of gene expression between TGIF and leptin). The FAMD indicated what variables account for the most variation in the dataset (with BMI > 25).

**Figure 2 cells-13-01240-f002:**
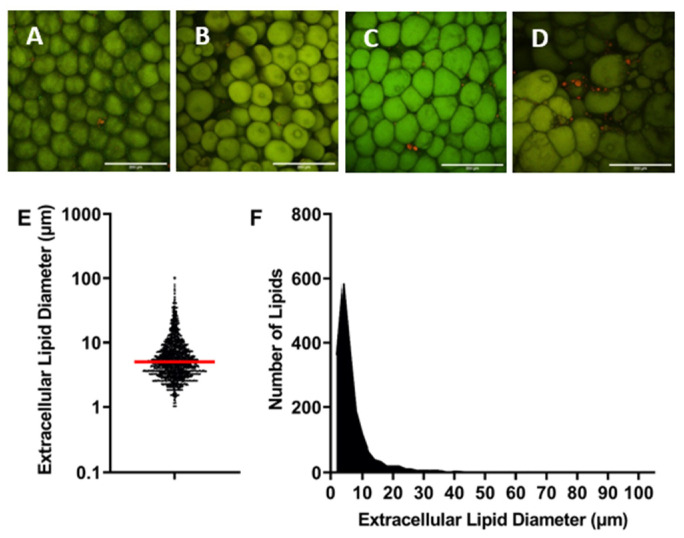
(**A**–**D**) Representative images used to analyze extracellular lipid diameter. Samples were stained with AdipoRed (red) and Phalloidin 488 (green). Extracellular lipids appear red, while adipocytes (stained by both AdipoRed and Phalloidin 488) appear green. Scale bars are 200 μm. Each image is from a different patient. (**E**) Each dot on the graph represents an extracellular lipid droplet diameter measurement, with the mean represented by the red line (mean—7.27 μm, standard deviation—8.16 μm). (**F**) Histogram showing frequency of extracellular lipids with specific diameters, where the majority of extracellular lipids measured were under 15 μm. *n* = 1904.

**Figure 3 cells-13-01240-f003:**
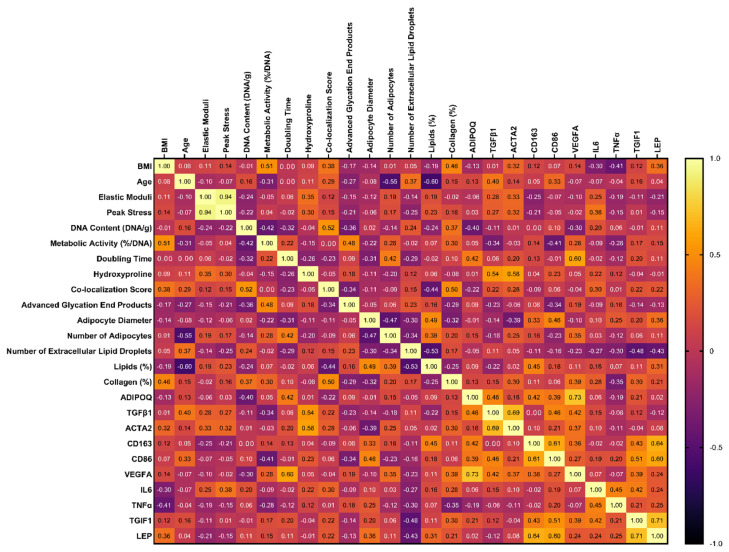
Amongst quantitative variables, many correlations exist in BMI, age, mechanical properties, cell number (DNA content), metabolic activity, doubling time of the stromal vascular fraction, collagen content (hydroxyproline), advanced glycation end products, morphological measurements, histological quantification, and gene expression. The correlation matrix indicates the correlation between quantitative variables (samples from N = 20 patients with BMI > 25). A value greater than 0.7 is considered a high correlation, 0.5–0.7 is considered a moderate correlation, and 0.3–0.5 is considered a low correlation.

**Figure 4 cells-13-01240-f004:**
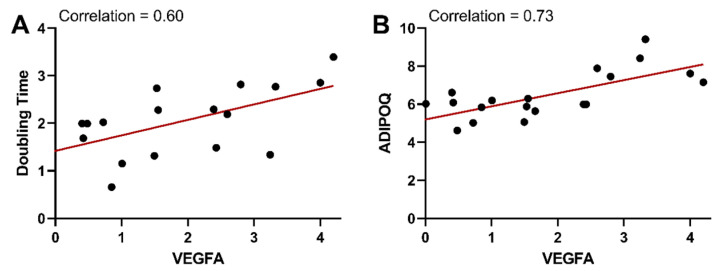
Gene expression of VEGFA was correlated to stromal vascular doubling time and adiponectin gene expression. VEGFA gene expression is plotted versus stromal vascular doubling time (**A**) and adiponectin (ADIPOQ) gene expression (**B**). Gene expression is represented as a delta CT from the housekeeping (HK) gene (ΔCT = CT_SDHA_ − CT_Target gene_). Therefore, the HK gene is equal to 0 on the plots. With the formula used, gene expression is relative to the housekeeping gene and increases at higher values. The red lines indicate a linear regression best fit line for each dataset. Correlations from the correlation matrix are indicated on each plot.

**Figure 5 cells-13-01240-f005:**
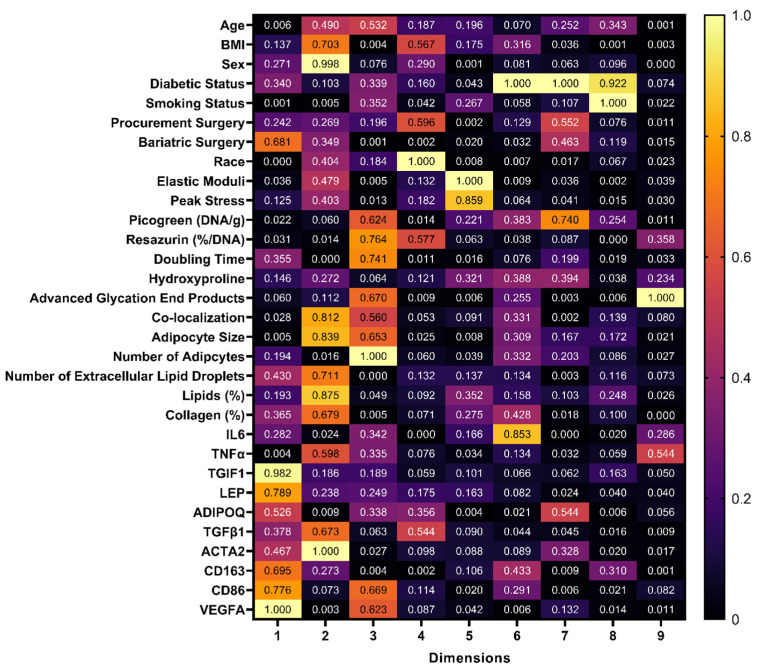
Contribution results generated from the FAMD analysis normalized to the highest contribution in each dimension. In each column, the contribution of variables to the nine dimensions is shown in a heat map, where a value of 1 indicates the largest loading of a variable and the highest contributor in that dimension. For example, in dimension 1, VEGFA is the highest contributor (1.00) followed closely by TGIF1 (0.982).

**Figure 6 cells-13-01240-f006:**
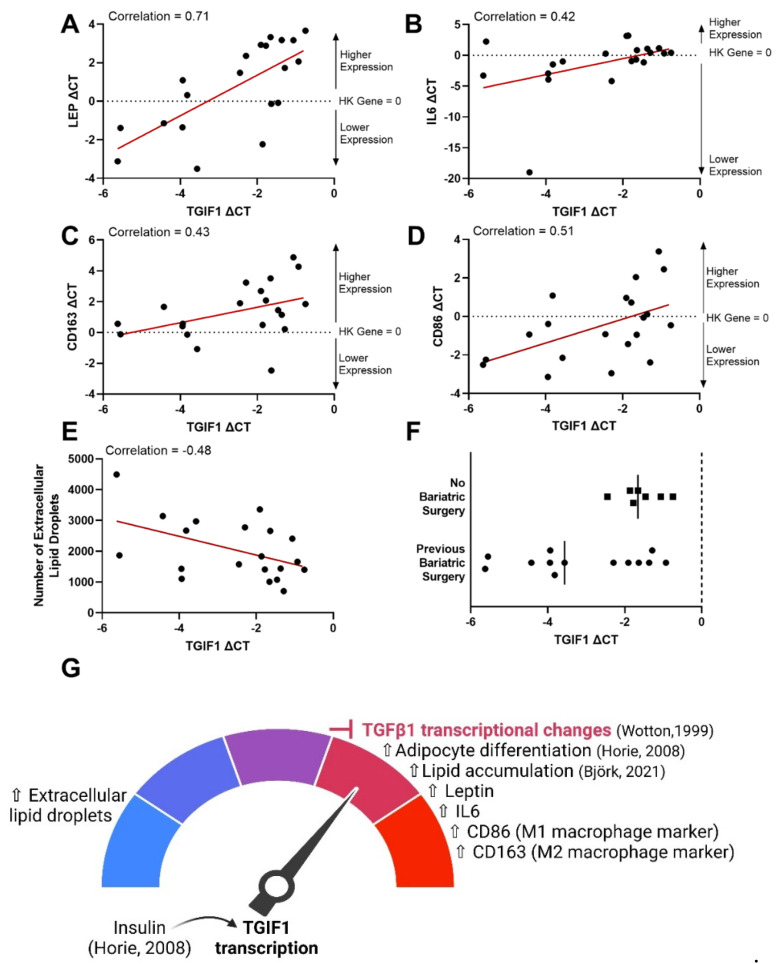
Increased gene expression of transforming growth factor-β induced factor homeobox 1 (TGIF1) was correlated with an increase in gene expression of leptin, interleukin-6 (IL6), CD163, CD86, and no bariatric surgery and a decrease in the number of extracellular lipid droplets. TGIF1 gene expression was plotted versus leptin (LEP) gene expression (**A**), IL6 gene expression (**B**), CD163 gene expression (**C**), CD86 gene expression (**D**), and the number of extracellular lipid droplets counted in histological images (**E**) and separated by whether the patient had undergone bariatric surgery previously or not (**F**). Gene expression is represented as a delta CT from the housekeeping (HK) gene (ΔCT = CT_SDHA_ − CT_Target gene_). Therefore, the HK gene is equal to 0 on the plots. With the formula used, gene expression is relative to the housekeeping gene and increases from a negative value to a higher positive value. The red lines indicate a linear regression best fit line for each dataset. Correlations from the correlation matrix are indicated on each plot. A summary of the results and current literature findings are illustrated (**G**), where the literature indicates that insulin upregulates TGIF1 (Horie, 2008) [60], blocking TGFβ1 transcriptional changes (Wotton, 1999) [57], inducing adipocyte differentiation (Horie, 2008) [60], and increasing lipid accumulation (Bjork, 2021) [61].

**Figure 7 cells-13-01240-f007:**
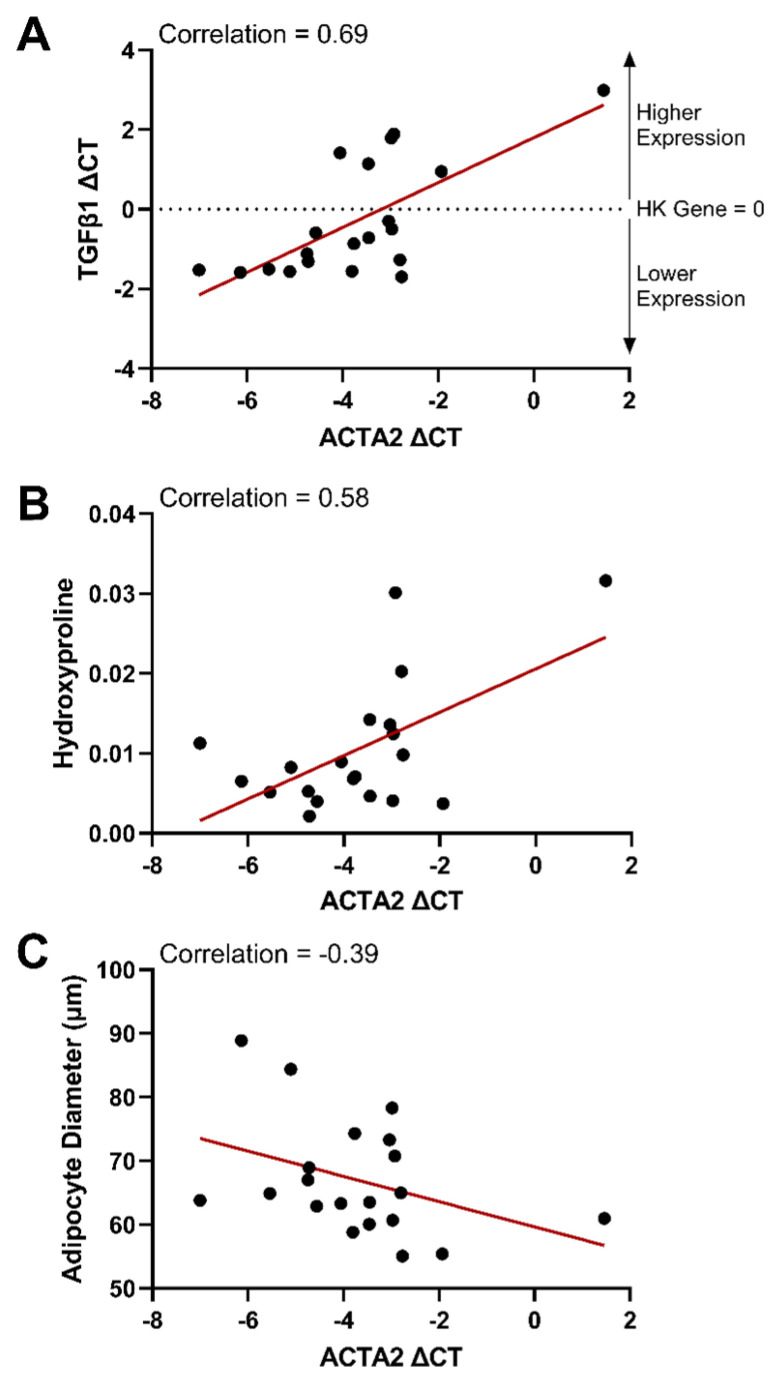
Increased gene expression of actin alpha 2, smooth muscle (ACTA2) gene expression was correlated with an increase in transforming growth factor-β (TGFβ1) gene expression and collagen content (hydroxyproline), with a corresponding decrease in adipocyte diameter. ACTA2 gene expression was plotted versus TGFβ1 (**A**), hydroxyproline content (**B**), and measurements of adipocyte diameter in histological images (**C**). Gene expression is represented as a delta CT from the housekeeping (HK) gene (ΔCT = CT_SDHA_ − CT_Target gene_). Therefore, the HK gene is equal to 0 on the plots. With the formula used, gene expression is relative to the housekeeping gene and increases from a negative value to a higher positive value. The red lines indicate a linear regression best fit line for each dataset. Correlations from the correlation matrix are indicated on each plot.

**Figure 8 cells-13-01240-f008:**
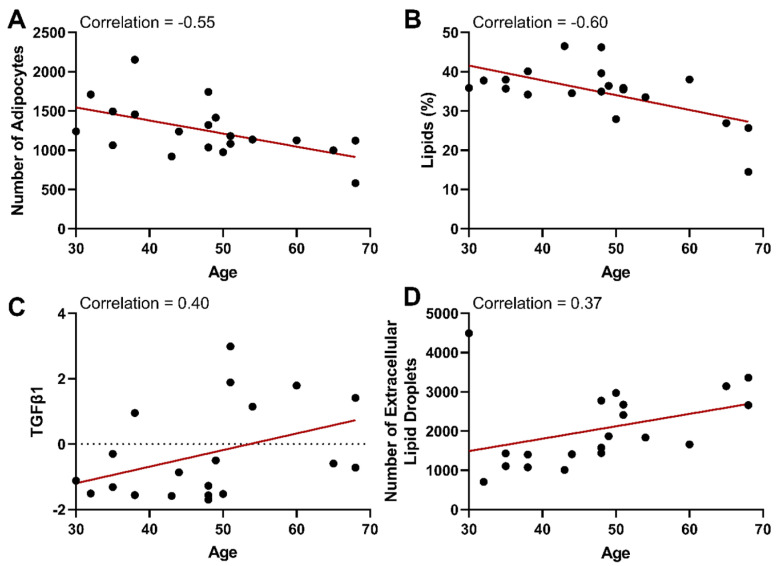
With increasing age, there is a decline in the number of adipocytes and the percentage of lipids and an increase in TGFβ1 and the number of lipid droplets deposited extracellularly. The age of patients at the time of surgery is plotted versus the number of adipocytes measured in histological images (**A**), the lipid percent measured in the same images (**B**), the gene expression of TGFβ1 (**C**), and the number of extracellular lipid droplets counted in histological images (**D**). TGFβ1 gene expression is represented as a delta CT from the housekeeping gene (ΔCT = CT_SDHA_ − CT_Target gene_). With the formula used, gene expression is relative to the housekeeping gene and increases from a negative value to a higher positive value. The red lines indicate a linear regression best fit line for each dataset. Correlations from the correlation matrix are indicated on each plot.

**Figure 9 cells-13-01240-f009:**
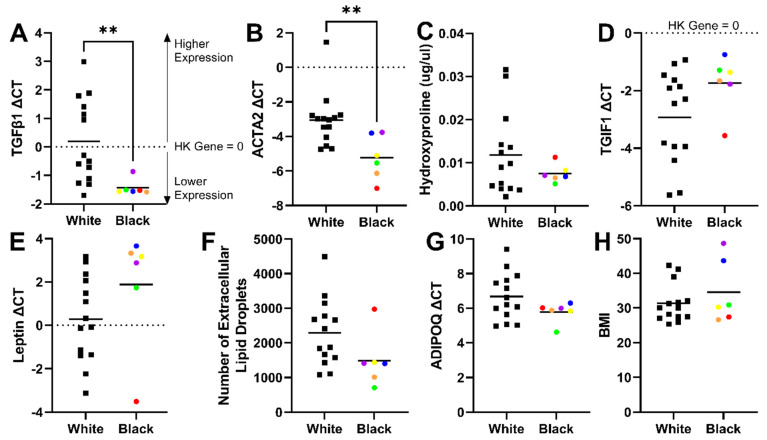
Significantly lower fibrotic gene expression (TGFβ1 and ACTA2) and trends related to collagen content, TGIF1, leptin, extracellular lipid droplets, and adiponectin were observed in self-identified Black patients compared to White patients and were independent of BMI. Samples derived from Black patients had significantly lower gene expression levels of TGFβ1 (**A**) and ACTA2 (**B**) and a trend towards lower hydroxyproline collagen content (**C**) than those derived from White patients (all patients identified as non-Hispanic). On average, samples from Black patients also had higher gene expression levels of TGIF1 (**D**) and leptin (**E**) and fewer extracellular lipid droplets (**F**) compared to White patients. Adiponectin levels were also lower in Black patient samples compared to White patient samples (**G**). Differences between the population groups are independent of body mass index (BMI), as there were no significant differences in BMI between the groups (**H**). Samples derived from Black patients are represented as dots and were color-coded by patient source (i.e., all of the blue dots are from the same patient) to highlight that the outliers in TGIF1, leptin, hydroxyproline, and number of extracellular lipid droplets were all from the same patient (red dots). The patient sample represented by the red dots follows the same trends as the aggregated data (with combined ancestries) where reduced levels of TGIF1 are linked with higher numbers of extracellular lipid droplets and an inability to suppress fibrotic changes in adipose tissue. Samples derived from White patients are represented as black squares. Black lines indicate the mean of the data. Statistical significance was determined by an unpaired *t*-test, where ** indicates *p* < 0.01.

**Figure 10 cells-13-01240-f010:**
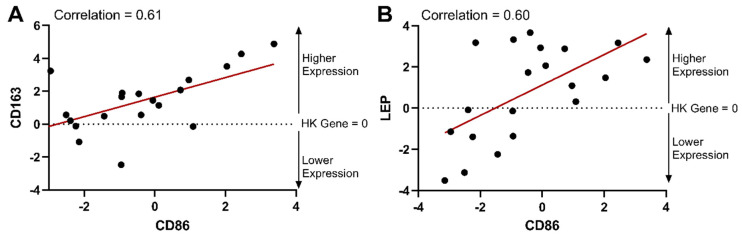
Gene expression of the classically activated CD86 macrophage surface marker and the alternatively activated CD163 surface marker were correlated with each other and with leptin gene expression. CD86 gene expression is plotted versus CD163 gene expression (**A**) and leptin (LEP) gene expression (**B**). Gene expression is represented as a delta CT from the housekeeping (HK) gene (ΔCT = CT_SDHA_ − CT_Target gene_). Therefore, the HK gene is equal to 0 on the plots. With the formula used, gene expression is relative to the housekeeping gene and increases from a negative value to a higher positive value. The red lines indicate a linear regression best fit line for each dataset. Correlations from the correlation matrix are indicated on each plot.

**Table 1 cells-13-01240-t001:** Breakdown of patient demographics used in this study.

Demographic		Number of Patients
**Race**	Black	6
	White	14
**Sex**	Female	17
	Male	3
**Surgery**	Abdominoplasty	5
	Panniculectomy	15
**Bariatric Surgery**	Previous Bariatric Surgery	13
	No Bariatric Surgery	7
**Diabetic Status**	Diabetic	5
	Formerly Diabetic	2
	Not Diabetic	13
**Smoking Status**	Non-Smoker	11
	Smoker	9

**Table 2 cells-13-01240-t002:** Full demographic information for patients used in this study.

Patient	Gender	BMI	Diabetic Status	Smoking Status	Age	Surgery	Weight Loss Method	Race
**1**	Female	27.06	Formerly Diabetic	Former Smoker	30	Panniculectomy	Gastric Band	White
**2**	Female	32.12	Not Diabetic	Non-Smoker	68	Panniculectomy	Roux-en-y	White
**3**	Female	25.92	Not Diabetic	Non-Smoker	54	Abdominoplasty	No Bariatric Surgery	White
**4**	Female	27.44	Not Diabetic	Former Smoker	49	Panniculectomy	Roux-en-y	White
**5**	Male	42.26	Diabetic	Former Smoker	51	Panniculectomy	Roux-en-y	White
**6**	Female	27.4	Not Diabetic	Non-Smoker	50	Panniculectomy	Gastric Band	Black
**7**	Female	31.6	Not Diabetic	Former Smoker	35	Panniculectomy	Gastric Sleeve	White
**8**	Female	39.02	Diabetic	Non-Smoker	65	Panniculectomy	Roux-en-y	White
**9**	Female	43.63	Diabetic	Non-Smoker	38	Panniculectomy	No Bariatric Surgery	Black
**10**	Male	41.21	Not Diabetic	Non-Smoker	48	Panniculectomy	Roux-en-y	White
**11**	Female	28.17	Formerly Diabetic	Non-Smoker	35	Abdominoplasty	Roux-en-y	White
**12**	Female	26.62	Not Diabetic	Former Smoker	43	Panniculectomy	No Bariatric Surgery	Black
**13**	Female	30.02	Pre-Diabetic	Former Smoker	51	Abdominoplasty	No Bariatric Surgery	White
**14**	Female	30.05	Not Diabetic	Former Smoker	68	Panniculectomy	Gastric Sleeve	White
**15**	Female	27.44	Diabetic	Non-Smoker	48	Abdominoplasty	No Bariatric Surgery	White
**16**	Female	30.23	Diabetic	Non-Smoker	48	Panniculectomy	Roux-en-y	Black
**17**	Female	31.31	Not Diabetic	Former Smoker	60	Panniculectomy	Gastric Sleeve	White
**18**	Female	25.34	Not Diabetic	Non-Smoker	38	Abdominoplasty	No Bariatric Surgery	White
**19**	Female	30.87	Not Diabetic	Smoker	32	Panniculectomy	Gastric Sleeve	Black
**20**	Male	48.66	Not Diabetic	Non-smoker	44	Panniculectomy	No Bariatric Surgery	Black

**Table 3 cells-13-01240-t003:** Eigenvalues and variance determined through factorial analysis of mixed data (FAMD) analysis.

*Dimension*	Eigenvalue	Variance (%)	Cumulative Variance (%)	Largest Loading
** *1* **	4.89	15.29	15.29	VEGFA
** *2* **	4.28	13.38	28.67	ACTA2
** *3* **	3.68	11.5	40.17	Number of Adipocytes
** *4* **	3.52	11.01	51.18	Race
** *5* **	3.05	9.52	60.7	Elastic Moduli
** *6* **	2.26	7.08	67.78	Diabetic Status
** *7* **	1.89	5.91	73.69	Diabetic Status
** *8* **	1.84	5.75	79.44	Smoking Status
** *9* **	1.53	4.78	84.22	Advanced Glycation End Products

## Data Availability

The raw data supporting the conclusions of this article will be made available by the authors on request.

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
