# Peer review of "A Preliminary Study on Factors That Drive Patient Variability in Human Subcutaneous Adipose Tissues"

_cells, 2024, doi:10.3390/cells13151240_

Round 1
Reviewer 1 Report
Comments and Suggestions for Authors
In this manuscript, the authors attempt to establish correlations and patterns between the biological, mechanical, and epidemiological properties of human adipose tissues. However, there are significant issues with the study as presented, including a superficial review of the literature, a low sample size, and the dismissal of genetic variation between individuals, which reduce the significance of the findings:
The introduction is superficial and requires substantial improvement to enhance the accuracy and depth of the information presented. For example, it is well-documented in the literature that fibrosis is a major determinant of the varying patient responses to bariatric surgery. Multiple studies have confirmed this finding, and scoring systems have been established (Divoux A et al., 2010; Bel Lassen P et al., 2017).
The sample size is insufficient to address the research questions posed by the authors. Additionally, the overlap of independent variables within each category is unclear. It is essential to determine whether the patient sample size is adequate for the analyses performed. For example, how many of the three male participants underwent bariatric surgery? What is the age distribution across sex and demographics? A comprehensive table detailing the characteristics of each patient would be beneficial. Increasing the sample size is imperative before drawing any conclusions.
The number of images analyzed per patient (2 per patient) is too low, limiting the potential for significant findings. The authors should consider using scanning techniques and automated analysis tools, such as ATAT (Robino JJ et al., 2024), to enhance accuracy and increase the number of images analyzed.
It is not clear how the different procedures for fat collection (abdominoplasty vs. panniculectomy) may affect the collected tissue. The authors should clarify whether these procedures influence cell expansion, the relative number of ASCs, and whether the collected cells and adipocytes are comparable.
The term "extra-cellular lipid droplet" needs further clarification. The authors should provide additional commentary on this observation and its significance.
The characterization of resident macrophages is overly simplistic, relying solely on the classical M1 and M2 paradigm. Limiting the observations to the expression of CD86 and CD163 is a significant shortcoming. The authors need to address this limitation by considering the complexity of macrophage populations (Strand K et al., 2022; Finlin BS et al., 2021).
Author Response
In this manuscript, the authors attempt to establish correlations and patterns between the biological, mechanical, and epidemiological properties of human adipose tissues. However, there are significant issues with the study as presented, including a superficial review of the literature, a low sample size, and the dismissal of genetic variation between individuals, which reduce the significance of the findings:
The introduction is superficial and requires substantial improvement to enhance the accuracy and depth of the information presented. For example, it is well-documented in the literature that fibrosis is a major determinant of the varying patient responses to bariatric surgery. Multiple studies have confirmed this finding, and scoring systems have been established (Divoux A et al., 2010; Bel Lassen P et al., 2017).
Thanks for bringing this up. We did keep the introduction concise as the paper is quite long with more depth and citations found throughout the results/discussion. We have added additional literature (including the recommended references) to enhance the depth of information in the introduction.
The sample size is insufficient to address the research questions posed by the authors. Additionally, the overlap of independent variables within each category is unclear. It is essential to determine whether the patient sample size is adequate for the analyses performed. For example, how many of the three male participants underwent bariatric surgery? What is the age distribution across sex and demographics? A comprehensive table detailing the characteristics of each patient would be beneficial. Increasing the sample size is imperative before drawing any conclusions.
Thank you for suggesting we add a comprehensive table detailing the characteristics of each patient, which is now Table 2. Of the three male patients included in this study 2/3 underwent Roux-en-y, 1 did not undergo bariatric surgery. The age distribution was 30-68. We also added additional language in the discussion on the need for increasing the sample size – “As is the case for most studies, this preliminary study of 20 patient samples would be strengthened by the addition of more patients and an increase in the diversity of the patient population.”
The number of images analyzed per patient (2 per patient) is too low, limiting the potential for significant findings. The authors should consider using scanning techniques and automated analysis tools, such as ATAT (Robino JJ et al., 2024), to enhance accuracy and increase the number of images analyzed.
Sorry for the confusion, there were actually 4 images taken per patient. In the text we have made this more clear by now stating:
“2 samples per patient were imaged, with 2 images being captured per sample, for a total of 4 images per patient and 80 images total for the whole study.”
Since we are doing z-stacks with our confocal, scanning techniques and automated tools are not available to us.
It is not clear how the different procedures for fat collection (abdominoplasty vs. panniculectomy) may affect the collected tissue. The authors should clarify whether these procedures influence cell expansion, the relative number of ASCs, and whether the collected cells and adipocytes are comparable.
There were no significant differences between abdominoplasty and pannuculectomy, as this was a variable included in the FAMD. The only significant findings related to surgeries were whether or not the patient had undergone bariatric surgery previously (which is mentioned where relevant throughout). We have added a line in the limitation section to make this clear –
“Although it should be noted that there were no significant differences between tissues obtained from abdominoplasty and panniculectomy surgeries.”
The term "extra-cellular lipid droplet" needs further clarification. The authors should provide additional commentary on this observation and its significance.
Thank you for this suggestion. We agree that the extracellular lipid droplets are significant and we have added more text on this subject in the conclusion to bring it into focus more –
Furthermore, this study draws attention to what we define as “extracellular lipid droplets” which were consistently found in the obese adipose tissues evaluated here. Reduced levels of TGIF1 were correlated with higher numbers of extracellular lipid droplets and an inability to suppress fibrotic changes in adipose tissue. We speculate there could be a metabolic phenotype in some patients that favors intracellular lipid accumulation with high TGIF1 transcription that limits extracellular lipid release. High values for the colocalization score signify that extracellular lipid droplets were primarily found in collagenous regions of the subcutaneous adipose tissues (consistent across all patients as seen in Supplemental Figure 1). Further work needs to define what initiates secretion/deposition of lipid droplets extracellularly and whether extracellular lipid droplets have a role and/or association with the progression of adipose tissue fibrosis.
The characterization of resident macrophages is overly simplistic, relying solely on the classical M1 and M2 paradigm. Limiting the observations to the expression of CD86 and CD163 is a significant shortcoming. The authors need to address this limitation by considering the complexity of macrophage populations (Strand K et al., 2022; Finlin BS et al., 2021).
We have added more text and literature citing the spectrum of macrophage polarization states as suggested, which we believe greatly improved the section – “Macrophage surface markers.”
Reviewer 2 Report
Comments and Suggestions for Authors
The authors investigated relatively large set of adipose tissue characteristics in 20 patients and correlated these characteristics with demographic parameters to assess the parameters responsible for SAT variability. The methodologies used and the overall data presentation and discussion are of high quality.
Perhaps the biggest limitation of this work is the number of subjects, which is really small for such a study focusing on AT variability. Which the authors themselves admit. If increasing the number of subjects is not possible, I would consider putting "preliminary or exploratory study“ in the title of the paper. At the same time the statement in the title that " human SAT variability is driven by VEGFA, ACTA2 etc".. seems relatively strong to me. Only a limited number of SAT characteristics have been studied in this paper, so it cannot be said that variability is indeed driven by these factors alone. I would choose a more moderate statement, such as "is associated".
Furthermore, it is not very clear to me how the authors chose the independent variables for the factorial and correlation analysis. The choice of 10 genes, in particular, is not substantially justified. How do the authors know that in this particular small set of selected genes will be the ones responsible for the greatest variability in AT? For this type of analysis, I would expect some previous more comprehensive methodology, e.g. microarrays or RNAseq.
On page 10, the authors discuss the proliferative capacity of ASCs. Did they also test differentiation capacity of the preadipocytes mentioned?
Is there any way to rule out that the data on variability in race is not dependent on FM? Unfortunately, BMI is not an entirely appropriate indicator of adiposity.
I have to admit that this is the first time I've heard of extracellular lipid droplets in SAT. Could the authors explain and discuss this parameter more? How are these droplets formed? Are they somehow related to the breakdown of adipocytes or what is their occurrence related to?
The finding of correlation of M1/M2 macrophage markers does not seem very "novel" to me. It has been shown previously that macrophages in human AT are not polarized into M1 and M2, but rather have a mixed phenotype (PMID: 30229891, PMID: 18227385).
Given that this is a study on human adipose tissue, it would be appropriate to compare/discuss more human studies.
P-value should be included in all correlation plots.
2-dCT or 2dCT (rather than dCt alone) is usually used to express the quantity of gene expression.
The legend in Figure 7 does not seem to correspond to the graphs shown (there is no D) and E) part in the figure).
Minor wording inaccuracies need to be corrected in the introduction: line 47-"…adipose tissue expansion in obesity is a risk factor for a plethora of diseases.." However, AT expansion is not always a risk factor, e.g. hyperplastic expansion is associated with heathy metabolic phenotype (as mentioned further as well). I would choose "accumulation of fat mass" or something similar; lines: 66-67, the phrasing „..adipocyte hypertrophy enables to store more lipids, protecting other organs from lipotoxic stress", seems confusing to me. Conversely, adipocyte hypertrophy has been associated with insufficient storage and lipotoxicity; line: 57 - " The predominant cell in AT is the adipocyte, which stores triglycerides that can be released into bloodstream" - however TAGs are not released from AT, only FA.
Author Response
The authors investigated relatively large set of adipose tissue characteristics in 20 patients and correlated these characteristics with demographic parameters to assess the parameters responsible for SAT variability. The methodologies used and the overall data presentation and discussion are of high quality.
Perhaps the biggest limitation of this work is the number of subjects, which is really small for such a study focusing on AT variability. Which the authors themselves admit. If increasing the number of subjects is not possible, I would consider putting "preliminary or exploratory study“ in the title of the paper. At the same time the statement in the title that " human SAT variability is driven by VEGFA, ACTA2 etc".. seems relatively strong to me. Only a limited number of SAT characteristics have been studied in this paper, so it cannot be said that variability is indeed driven by these factors alone. I would choose a more moderate statement, such as "is associated".
Thank you for this suggestion, we have adjusted the title accordingly to:
“A preliminary study on factors that drive patient variability in human subcutaneous adipose tissues”
and believe it does a much better job of describing the study.
Furthermore, it is not very clear to me how the authors chose the independent variables for the factorial and correlation analysis. The choice of 10 genes, in particular, is not substantially justified. How do the authors know that in this particular small set of selected genes will be the ones responsible for the greatest variability in AT? For this type of analysis, I would expect some previous more comprehensive methodology, e.g. microarrays or RNAseq.
We used the literature to justify the gene choices as we have motivated in the text, and unfortunately did not have the budget to do microarray or RNAseq. However, we do plan on following up with studies with these techniques in the future if funding allows.
On page 10, the authors discuss the proliferative capacity of ASCs. Did they also test differentiation capacity of the preadipocytes mentioned?
Unfortunately, we did not test the differentiation capacity of the ASCs and have added this as a limitation in our discussion.
Is there any way to rule out that the data on variability in race is not dependent on FM (fat mass)? Unfortunately, BMI is not an entirely appropriate indicator of adiposity.
We agree that BMI is not an accurate indicator of adiposity. We do not have any information on fat mass and have added this as a limitation to the text.
I have to admit that this is the first time I've heard of extracellular lipid droplets in SAT. Could the authors explain and discuss this parameter more? How are these droplets formed? Are they somehow related to the breakdown of adipocytes or what is their occurrence related to?
We haven’t found any literature on extracellular lipid droplets in SAT either, although we consistently see them in all of our samples (to varying degrees) which motivated the measurement of this parameter in this study. We are following up with studies on this topic! We think this finding is very interesting and hope to have another paper published on this finding in the near future. We have added more discussion of this parameter in the conclusion –
Furthermore, this study draws attention to what we define as “extracellular lipid droplets” which were consistently found in the obese adipose tissues evaluated here. Reduced levels of TGIF1 were correlated with higher numbers of extracellular lipid droplets and an inability to suppress fibrotic changes in adipose tissue. We speculate there could be a metabolic phenotype in some patients that favors intracellular lipid accumulation with high TGIF1 transcription that limits extracellular lipid release. High values for the colocalization score signify that extracellular lipid droplets were primarily found in collagenous regions of the subcutaneous adipose tissues (consistent across all patients as seen in Supplemental Figure 1). Further work needs to define what initiates secretion/deposition of lipid droplets extracellularly and whether extracellular lipid droplets have a role and/or association with the progression of adipose tissue fibrosis.
The finding of correlation of M1/M2 macrophage markers does not seem very "novel" to me. It has been shown previously that macrophages in human AT are not polarized into M1 and M2, but rather have a mixed phenotype (PMID: 30229891, PMID: 18227385).
Thank you for pointing this out. We have adjusted the text to support current literature and weren’t trying to imply this finding was novel, but rather supportive of limited literature on this subject. We hope the new text reflects this better. We have also included the additional suggested citations in our discussion.
Given that this is a study on human adipose tissue, it would be appropriate to compare/discuss more human studies.
We have included more literature on human adipose tissue studies where we could.
P-value should be included in all correlation plots.
Thank you for this suggestion. We have included the correlation value on every plot as this is more relevant for the data presented.
2-dCT or 2dCT (rather than dCt alone) is usually used to express the quantity of gene expression.
We agree that when there is a control it is best to represent data with the delta-delta Ct method (2–∆∆Ct). Since we do not have a treatment versus control comparison we have just represented the delta CT from the housekeeping gene.
The legend in Figure 7 does not seem to correspond to the graphs shown (there is no D) and E) part in the figure).
Thank you for catching this. We have updated the legend to match the figure.
Minor wording inaccuracies need to be corrected in the introduction: line 47-"…adipose tissue expansion in obesity is a risk factor for a plethora of diseases.." However, AT expansion is not always a risk factor, e.g. hyperplastic expansion is associated with heathy metabolic phenotype (as mentioned further as well). I would choose "accumulation of fat mass" or something similar; lines: 66-67, the phrasing „..adipocyte hypertrophy enables to store more lipids, protecting other organs from lipotoxic stress", seems confusing to me. Conversely, adipocyte hypertrophy has been associated with insufficient storage and lipotoxicity; line: 57 - " The predominant cell in AT is the adipocyte, which stores triglycerides that can be released into bloodstream" - however TAGs are not released from AT, only FA.
Thank you for pointing out these inaccuracies. We have adjusted them as suggested.
Round 2
Reviewer 1 Report
Comments and Suggestions for Authors
The revised manuscript and the authors' responses are clearer. However, before I can agree to accept the paper, I would like to see a staining of the extracellular lipid droplets with a stain that specifically delineates the adipocytes. This is crucial to support the idea that extracellular lipid droplets accumulate outside of adipocytes, which is a novel and exciting finding.
Author Response
Thank you for your continued interest in these extracellular lipid droplets! Figure 2 demonstrates that the lipid droplets are extracellular as they only stain positive for Adipored and are not counter stained with phalloidin (which stains the actin cytoskeleton). Adipocytes stain positive for both Adipored and Phalloidin 488 and appear yellowish (red + green).
Round 3
Reviewer 1 Report
Comments and Suggestions for Authors
Thank you for addressing the comments and feedback. I find your responses satisfactory and the revisions have significantly improved the manuscript.